# DNA Double-Strand Break Repair Inhibitors: YU238259, A12B4C3 and DDRI-18 Overcome the Cisplatin Resistance in Human Ovarian Cancer Cells, but Not under Hypoxia Conditions

Anna Macieja [1] , Izabela Gulbas [2] and Tomasz Popławski [1],*

1 Department of Microbiology and Pharmaceutical Biochemistry, Medical University of Lodz, Mazowiecka 5, 92-215 Lodz, Poland; anna.macieja@umed.lodz.pl
2 Department of Immunology and Allergy, Medical University of Lodz, Pomorska 251, 92-213 Lodz, Poland; izabela.gulbas@umed.lodz.pl
* Correspondence: tomasz.poplawski@umed.lodz.pl

**Abstract:** Cisplatin (CDDP) is the cornerstone of standard treatment for ovarian cancer. However, the resistance of ovarian cancer cells to CDDP leads to an inevitable recurrence. One of the strategies to overcome resistance to CDDP is the combined treatment of ovarian cancer with CDDP and etoposide (VP-16), although this strategy is not always effective. This article presents a new approach to sensitize CDDP-resistant human ovarian carcinoma cells to combined treatment with CDDP and VP-16. To replicate the tumor conditions of cancers, we performed analysis under hypoxia conditions. Since CDDP and VP-16 induce DNA double-strand breaks (DSB), we introduce DSB repair inhibitors to the treatment scheme. We used novel HRR and NHEJ inhibitors: YU238259 inhibits the HRR pathway, and DDRI-18 and A12B4C3 act as NHEJ inhibitors. All inhibitors enhanced the therapeutic effect of the CDDP/VP-16 treatment scheme and allowed a decrease in the effective dose of CDDP/VP16. Inhibition of HRR or NHEJ decreased survival and increased DNA damage level, increased the amount of γ-H2AX foci, and caused an increase in apoptotic fraction after treatment with CDDP/VP16. Furthermore, delayed repair of DSBs was detected in HRR- or NHEJ-inhibited cells. This favorable outcome was altered under hypoxia, during which alternation at the transcriptome level of the transcriptome in cells cultured under hypoxia compared to aerobic conditions. These changes suggest that it is likely that other than classical DSB repair systems are activated in cancer cells during hypoxia. Our study suggests that the introduction of DSB inhibitors may improve the effectiveness of commonly used ovarian cancer treatment, and HRR, as well as NHEJ, is an attractive therapeutic target for overcoming the resistance to CDDP resistance of ovarian cancer cells. However, a hypoxia-mediated decrease in response to our scheme of treatment was observed.

**Keywords:** ovarian cancer; YU238259; A12B4C3; DDRI-18; cisplatin; DNA double strand breaks

## 1. Introduction

Ovarian cancer is one of the most common malignancies among women, and it has been responsible for 5% of cancer-related deaths among them [1,2]. In western and central Europe, ovarian cancer incidence and mortality rates of ovarian cancer are relatively stable at a stage of 10.7 per 100,000 and 5.6 per 100,000, respectively [2]. Ovarian cancer detected at an early stage is characterized by an excellent prognosis (more than 90% of the 5-year survival rate) [3]. However, due to the early asymptomatic phase of the disease, most cases are detected at the advanced stages. It dramatically decreases the effectiveness of therapy [4,5].

For years, among the crucial drugs used for the cure of newly diagnosed ovarian cancer, paclitaxel and cyclophosphamide combined with platinum-containing compounds were CDDP or carboplatin. The progression occurs in approximately 80% of cases among women with the disease diagnosed in an advanced stage [6,7]. Unfortunately, recurrent

ovarian cancer is often characterized by the development of resistance to first-line treatment. A second-line treatment used after the development involves liposomal doxorubicin (acts as an inhibitor of replication), gemcitabine (replaces cytidine during replication and inhibits tumor growth), topotecan (topoisomerase I inhibitor), and VP-16 (topoisomerase II "poison") [8]. The "van der Burg protocol" (VDB) is one of the standardized treatment schemes for recurrent cisplatin-resistant ovarian cancer. This protocol involved the administration of CDDP and VP-16 and was initially described by patients as highly effective and well-tolerated [9]. However, recent studies revealed some severe limitations of this scheme treatment. High toxicity leading to numerous side effects (fatigue, nausea, vomiting) and even causes of death were observed among women treated with the VDB protocol [10].

One of the possible ways to overcome or partially minimize these limitations may be to use the treatment scheme compounds that will sensitize ovarian cancer cells to drugs used in the VDB protocol. It may help reduce the drug-induced toxicity. The mechanism of the components of the action of VDB protocol (CDDP and VP-16) involves the formation directly and indirectly of one of the most harmful types of DNA damage, double-strand breaks (DSBs). Therefore, our concept includes the introduction of small-molecule DSB repair inhibitors (DRIs) into the treatment scheme. DSB repair in human cells is carried out mainly by two clausal pathways: nonhomologous end joining (NHEJ) and homologous recombination (HRR). However, there are also distinguished auxiliary systems based on microhomology, such as microhomology-mediated end joining (MMEJ). We decided to use in our studies three DRIs, YU238259, A12B4C3, and DDRI-18, since HRR and NHEJ are responsible for repairing the DSBs evoked using CDDP/VP-16 treatment. The first is described as a potent inhibitor of the HRR pathway with an unknown mechanism of action [11]. The second is known as an inhibitor of the NHEJ pathway with hPNKP as a target molecule [12]. The third also inhibits the NHEJ pathway, but its mechanism of action is still unknown [13].

The present study is expected to contribute to our understanding of the molecular mechanism of combined treatment with DRI in ovarian cells. This approach is original in its assumptions and may help overcome the severe side effects of second-line treatment by decreasing the effective dose of CDDP and VP-16. This is the first work to propose increasing the effectiveness of anticancer treatment using DRI. The results obtained in the present study suggest that DSB inhibitors could be considered for further research on the treatment of ovarian cancer, but this approach is limited due to the limiting role of hypoxia.

## 2. Materials and Methods

### 2.1. Cell Lines

A2780, A2780cis, PEA1, and PEA2 cells representing human ovarian carcinoma were obtained from the European Collection of Authenticated Cell Cultures (Sigma Aldrich, St. Louis, MO, USA). Cells were divided into two experimental schemes (ES 1 and ES 2), each of them consisting of cisplatin-sensitive and cisplatin-resistant cell lines. Preliminary results were obtained using the experimental scheme ES 1 consisting of the cisplatin-sensitive A2780 cell line and its cisplatin-resistant counterpart, A2780cis [14]. ES 1 was chosen for preliminary verification of the hypothesis of this study. The variant of A2780cis is commonly used in combination with A2780 to study the phenomenon of drug resistance in ovarian cancer [14,15]. However, these cell lines have some serious disadvantages; that is, the A2780cis cell line was created in vitro by exposure of the parental A2780 line to increasing concentrations of CDDP [16]. The limitations of ES 1 cells are described in detail in the Discussion. Therefore, we decided to use PEA1/PEA2 cells that were obtained from a patient diagnosed with ovarian cancer at different stages of treatment. The characteristics of the cell lines are shown in Table 1 and Figure 1. Microscopic images of the cell lines are presented in the Supplementary Materials (Figure S1).

**Table 1.** Characteristics of ovarian cancer cell lines.

| | Experimental Scheme 1 (ES 1) | |
|---|---|---|
| **Cell line** | A2780 | A2780cis |
| **ECACC number** | 93112519 | 93112517 |
| **Description** | Cell lines established from previously untreated patients [17]. | Cell line obtained in vitro after treating parent A2780 cells with increasing concentrations of CDDP [14]. |
| **Morphology** | Epithelial cells | Epithelial cells |
| **Growth mode** | Adherent | Adherent |
| **Doubling time (hours)** | 18 | 20 |
| **Sensitivity to CDDP** | Sensitive | Resistant |
| | Experimental Scheme 2 (ES 2) | |
| **Cell line** | PEA1 | PEA2 |
| **ECACC number** | 10032306 | 10032307 |
| **Description** | PEA1 was collected prior to treatment with CDDP and prednimustine. Human ovarian cancer, estrogen positive [18]. | Cells collected from the same patient as PEA1 after treatment with CDDP and prednimustine [19]. |
| **Morphology** | Epithelial; swirling pattern of cells | Epithelial, swirling pattern of cells |
| **Growth mode** | Adherent | Adherent |
| **Doubling time (hours)** | 37 | 66 |
| **Sensitivity to CDDP** | Sensitive | Resistant |

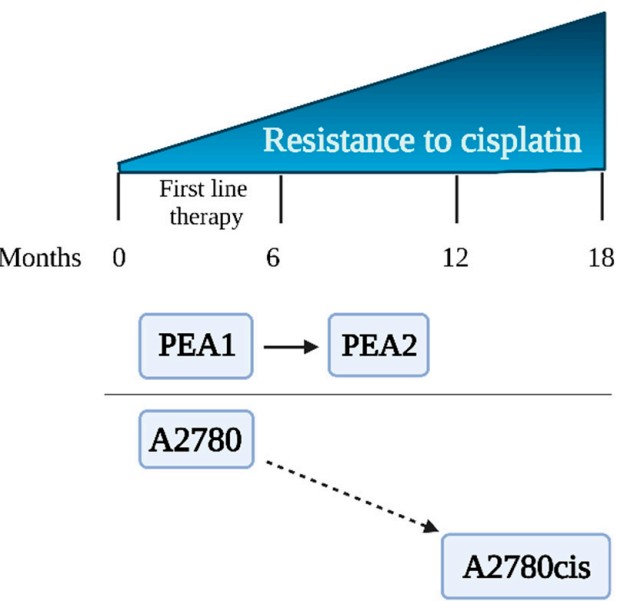

**Figure 1.** Ovarian cancer cell lines—sensitivity to cisplatin.

*2.2. Chemicals*

Drugs: VP-16, CDDP, and two inhibitors, A12B4C3 and DDRI-18, were obtained from Sigma-Aldrich (St. Louis, MO, USA), while the third inhibitor—YU238259—was synthesized using TriMen Chemicals (Lodz, Poland) as previously described [11]. VP-16 was dissolved in methanol in the 10 mM stock solution. CDDP was dissolved in water in the 5 mM stock solution. A12B4C3, DDRI-18, and YU238259 were prepared as 50 mM stock solutions in anhydrous dimethylsulfoxide (DMSO). The chemical structure of the drugs and inhibitors is shown in Figure 2. Other chemicals: Normal melting point (NMP) and low

melting point (LMP) agarose, Triton X-100, Tris HCl, phosphate-buffered saline (PBS), NaCl, NaHCO3, NaOH, ethylenediaminetetraacetic acid (EDTA), 4′,6-diamidino-2-phenylindole (DAPI), propidium iodide (PI), DNase-free RNAse were obtained from Sigma-Aldrich (St. Louis, MO, USA).

**Figure 2.** Chemical structures of drugs and inhibitors used in the study. (**A**) CDDP (cisplatin), (**B**) VP-16 (etoposide), (**C**) YU238259, (**D**) A12B4C3, (**E**) DDRI-18.

### 2.3. Cell Culture in Normoxia and Hypoxia Conditions

Cells were cultured in RPMI1640 medium with 2 mM L-glutamine obtained from Biological Industries, supplemented with 10% ($v/v$) fetal bovine serum (FBS) and 1% ($v/v$) penicillin/streptomycin from Corning (Tewksbury, MA, USA). For the PEA1 and PEA2 cell lines, the medium was additionally supplemented with 2 mM sodium pyruvate (Sigma-Aldrich, St. Louis, MO, USA). For cisplatin-resistant lines (A2780cis and PEA2), 1 μM CDDP (Sigma-Aldrich, St. Louis, MO, USA) was added to the culture medium at two passage intervals to maintain resistance to cisplatin. In standard cell culture, the oxygen level described as normoxia is equal to 18.6% (132.5 mmHg) of $O_2$ in the presence of atmospheric air (21% volume fraction of $O_2$) and 5% $CO_2$ in the incubator [18]. An oxygen level of less than 21% in standard cell culture is caused using water vapor and the addition of partial pressures of $CO_2$; it was described in detail by Place et al. [20]. In the present work, cells were cultured in the presence of atmospheric air enriched by 5% $CO_2$ at 37 °C in a humidified atmosphere to provide normoxia conditions. Hypoxia in cell culture is defined as insufficient, below the standard oxygen level. To provide hypoxia conditions, cells were cultured using the reagents described above in a hypoxic chamber (STEMCELL Technologies Germany GmbH, Köln, Germany). Cell culture was maintained in the presence of a gas mixture composed of 5% $CO_2$, 94.5% $N_2$, and 0.5% $O_2$. According to the manufacturer's instructions, the hypoxic chamber was filled with a mixture of gases each time before the start of incubation for 5 min, with a gas flow rate of 20 L/min. The flow velocity was measured with a manufacturer-recommended flowmeter (STEMCELL Technologies Germany GmbH, Köln, Germany). The chamber with culture vessels was

closed and placed in an incubator at 37 °C and a relative humidity of 90–95% after filling with the gas mixture. The appropriate level of humidity was provided by placing a Petri dish with distilled water in the hypoxic chamber.

### 2.4. Viability Assay

Cells were seeded in 96-well plates at the density of $2 \times 10^4$/well for ES 1 and $10^4$/well for ES 2 in 100 µL medium and incubated for at least 24 h until attached to the surface. Drugs: CDDP and VP-16 were added to the cells for 48 h immediately or after the pretreatment of cells with DRIs. Untreated control, positive control, and control with solvents used for stocks were performed (at least three replicates). Growth inhibition was assessed with a colorimetric Cell Counting 8 (CCK-8) kit (Sigma Aldrich, Poznań, Poland). The results were measured on a plate reader ($\lambda = 450$ nm) and presented as the percentage of untreated control. The concentration that induced 50% growth inhibition ($IC_{50}$) was calculated with Compusyn software, version 1.0 [21]. The doses of drugs for combined treatment were calculated based on $IC_{50}$ values obtained for CDDP and VP-16. The drug ratio was constant (1:1) and was determined according to the published recommendations for in vitro testing of anticancer compounds. We decided to use concentrations equal to $1/8$ $IC_{50}$, $1/4$ $IC_{50}$, $1/2$ $IC_{50}$, $IC_{50}$, $2$ $IC_{50}$, and $4$ $IC_{50}$ [21].

### 2.5. Comparison of the Influence of DRIs on the Effect of CDDP and VP-16

Cells were pretreated with DRI: 10 µM YU238259 or DDRI-18 for 60 min or 5 µM of A12B4C3 for 120 min. Subsequently, cells were treated with CDDP and VP-16 as single drugs and in combination. Pretreatment with DRI allowed us to determine whether these compounds can sensitize ovarian cancer cells to anticancer drugs. Cells were pretreated with YU238259 or DDRI-18 for 60 min or with A12B4C3 for 120 min before adding a mixture of CDDP/VP-16. Our preliminary results indicated that with YU238259 and DDRI-18, the effect of sensitization can be achieved after 60 min of pretreatment, while the use of A12B4C3 requires at least 120 min of pretreatment. It may be the result of differences in the cellular uptake of these compounds. A reduction factor value (Rf) was used as an indicator of chemosensitization. Rf was calculated as the ratio of $IC_{50}$ obtained after treatment with the drug(s) without inhibitor to the $IC_{50}$ obtained after pretreatment with each of the DRIs. The Rf value >1 shows chemosensitization. The ratio of CDDP and VP-16 was based on the $IC_{50}$ values in the combined treatment and was constant. The exact type of interactions (from synergism to antagonism) between the compounds tested was assessed using the combination index (CI). The $IC_{50}$ values determined for CDDP/VP-16-treated cells with or without pretreatment with DRIs served as a source for further CI calculations. CI < 1, CI = 1, and CI > 1 indicate synergistic, additive, or antagonistic interactions, respectively. CI was calculated with Compusyn software [21] and is described in more detail in Section 2.13.

### 2.6. Comet Assay

An alkaline comet assay was performed to assess the drug-induced level of DSBs and to investigate the process of DSB repair. The comet assay was performed as previously described [22], with modifications. To measure the level of DNA damage, cells were exposed for 2 h to the compounds. The DNA repair process was observed when the drug-treated cells were centrifuged and then placed in a drug-free medium. We evaluated the influence of DRIs on the genotoxic activity of CDDP/VP-16 at the range of concentrations equal to $1/4$ $IC_{50}$ + $1/4$ $IC_{50}$, $1/2$ $IC_{50}$ + $1/2$ $IC_{50}$, $1$ $IC_{50}$ + $1$ $IC_{50}$ for ES 1 (Figure S5A,B) and $1/8$ $IC_{50}$ + $1/8$ $IC_{50}$, $1/4$ $IC_{50}$ + $1/4$ $IC_{50}$, $1/2$ $IC_{50}$ + $1/2$ $IC_{50}$ for ES 2 (Figure 3A,B). Cells were preincubated with 10 µM of DDRI-18 for 60 min and 5 µM of A12B4C3 for 120 min before 6 h of drug treatment. For each time point, cells were suspended in 0.75% LMP agarose and then placed on microscope slides precoated with 0.5% LMP agarose. The slides were placed in lysis buffer (2.5 M NaCl, 100 mM EDTA, 10 mM Tris, and 1% (*v/v*) Triton X-100) and incubated overnight at 4 °C. The slides were then moved for 20 min, the unwinding buffer appropriate for the alkaline version of the comet assay. An electrophoresis was performed

(17 V, 32 mA, 20 min), and the slides were dried and prepared for analysis using staining with DAPI (5 μg/mL) (Sigma Aldrich, Poznań, Poland).

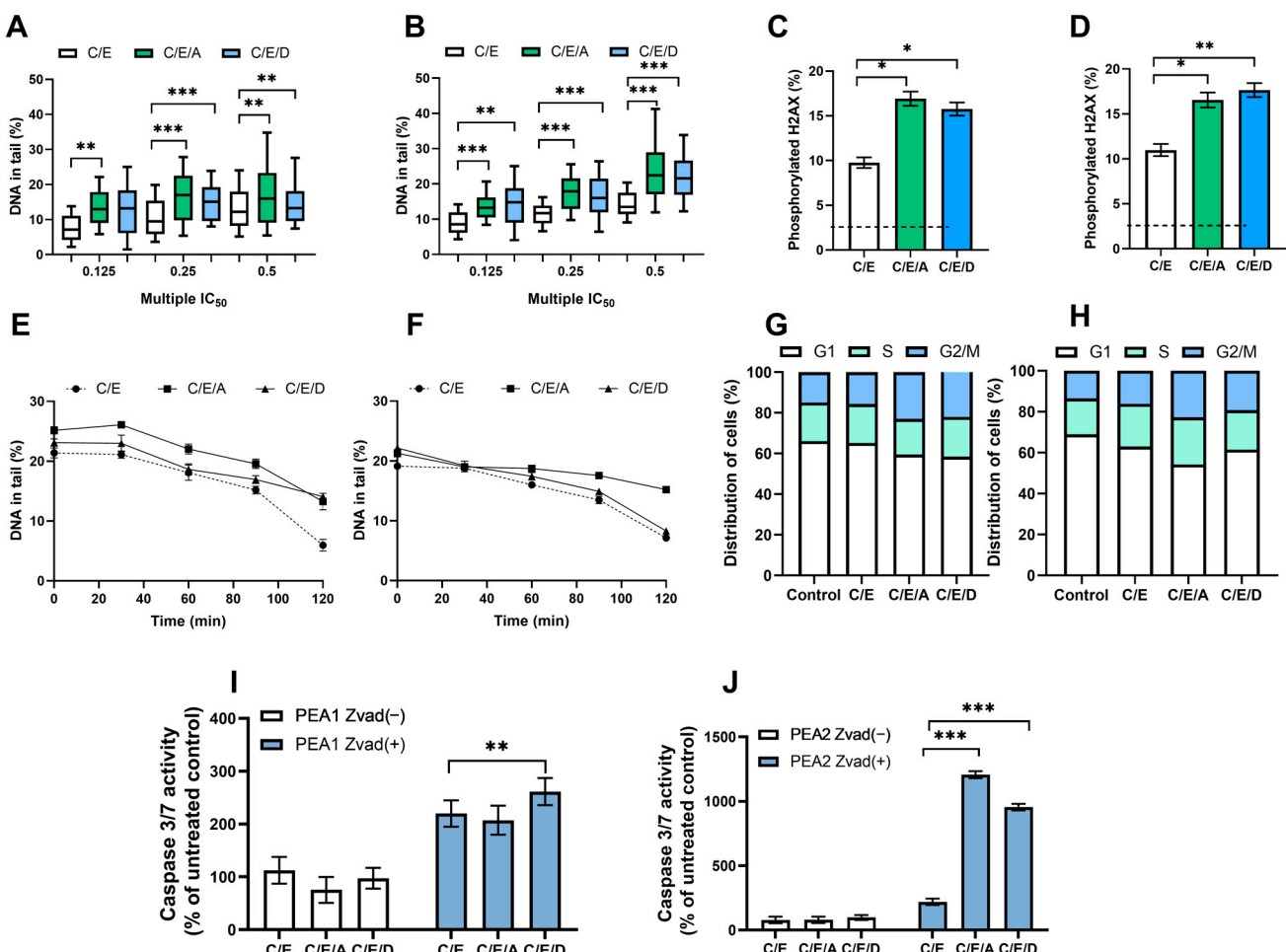

**Figure 3.** In normoxia conditions, DRIs are able to improve the activity of CDDP/VP-16 combination against human ovarian cancer cells PEA1 (**A**,**C**,**E**,**G**,**I**) and PEA2 (**B**,**D**,**F**,**H**,**J**). DRIs significantly increase CDDP/VP-16 induced level of DNA damage (**A**,**B**), the level of phosphorylated H2AX (**C**,**D**), modulate DNA repair (**E**,**F**), affect the distribution of cell cycle (**G**,**H**) and caspases 3 and 7 activity (**I**,**J**). Results are presented as the mean ± SEM (**A**,**B**) or mean ± SD (**C**–**J**) of 3 independent experiments, *—$p < 0.05$; **—$p < 0.01$; ***—$p < 0.001$.

### 2.7. H2AX Histone Phosphorylation

We used a commercially available kit, Human H2A.X (phospho S139), in cell ELISA Kit (IR) (ab131382) (Abcam, Cambridge, UK). It allows the phosphorylated Ser 193 in H2AX, using A rabbit monoclonal antibody specific for H2A.X phospho S139. The experiments were carried out according to the manufacturer's instructions. The treatment scheme was identical to that described in the previous section.

### 2.8. DNA Damage and Repair Analysis

The comet assay analysis was performed with an Eclipse fluorescence microscope (Nikon, Tokyo, Japan) attached to a COHU 4910 video camera (Cohu, Poway, CA, USA), using the computer-based image analysis system, Lucia Comet 4.51 (Laboratory Imaging, Praha, Czech Republic). The microscope is equipped with a UV filter block consisting of an excitation filter (λ = 359 nm) and a barrier filter (λ = 461 nm). DNA repair efficiency was measured after 60 min incubation with 10 μM DDRI-18 or 120 min of incubation with 10 μM A12B4C3 and another 2 h exposure to CDDP and VP-16 alone and in combination.

The DNA damage repair process was evaluated after 120 min of incubation in a drug-free medium. After that, cells were collected at time points from 0 to 2 h. As a time 0 (the initial level of DNA damage) we used cells collected directly after drug treatment. The level of DNA damage from the remaining time points was compared with that obtained for time 0. The level of DNA damage was measured as the percent of DNA in the comet tail.

### 2.9. Cell Cycle Analysis

The effect of sensitization with DDRI-18 and A12B4C3 on the cell cycle distribution was determined in CDDP-sensitive and CDDP-resistant ovarian cancer cells. DNA content related to the cell cycle phase was analyzed with an LSRII flow cytometer (Becton Dickinson, East Rutherford, NJ, USA). We compared cells treated with CDDP/VP-16 with those sensitized with DRIs before treatment with drugs. The sensitization of cells was carried out as described in previous sections. After 48 h of exposure to test compounds, samples were fixed with 96% ethanol and stained with PI (40 µg/mL) and RNase (200 µg/mL) for 30 min at 37 °C. For each experiment, positive control with 10 µM nocodazole, negative control (untreated cells), and unstained control samples were prepared. Differences in the DNA content allowed us to distinguish a percentage of cells in each cell cycle phase.

### 2.10. Detection of Apoptosis

Apoptosis was detected with the Apo-ONE® homogeneous caspase-3/7 assay (Promega, Walldorf, Germany), according to the manufacturer's instructions. It allows the detection of the active form of caspases 3 and 7—effector caspases in apoptosis. Cells were exposed to 10 µM of DDRI-18 or 5 µM of A12B4C3 before 24 h of treatment with CDDP/VP-16. The assay contains a profluorescent caspase-3/7 substrate, rhodamine 110 bis-(N-CBZ-l-aspartyl-l-glutamyl-l-valyl-aspartic acid amide) (Z-DEVD- R110), which becomes fluorescent after cleavage using caspase-3/7 enzymes. The amount of fluorescent product indicates the level of active caspase-3/7 in the sample. For each experiment, positive control cells were exposed to 100 µM VP-16, and negative control samples were performed. Cells were exposed to CDDP and VP-16 as single drugs or in combined treatment with or without DRI. Furthermore, we included the experimental scheme 20 µM pan-caspase inhibitor carbobenzoxy-valyl-alanyl-aspartyl-[O-methyl]-fluoromethylketone (Z-VAD-FMK) (Promega, Germany). It allowed confirming the apoptosis-mediated activation of caspases 3/7. The activity of 3/7 caspases was defined as a % activity of caspases 3/7 in the sample, compared to the untreated control.

### 2.11. Measurement of HIF1A Level from Total Protein

The Pierce TM BCA Protein Assay Kit (Thermo Fisher Scientific, Waltham, MA, USA) was used to determine the total protein level. Protein level was determined colorimetrically using a reagent containing BCA (bicinchoninic acid). Absorbance was measured at $\lambda$ = 562 nm using a Synergy TM HTX reader (BioTek, Winooski, VT, USA). The HIF1A Human ELISA Kit (Thermo Fisher Scientific, Waltham, MA, USA) was used to determine the HIF1A level. Markings were made according to the manufacturer's instructions. The absorbance measurement was carried out at wavelength $\lambda$1 = 450 nm and $\lambda$2 = 550 nm using the SynergyTM HTX reader (BioTek, Winooski, VT, USA).

### 2.12. Gene Expression

The RNeasy Mini Kit (Qiagen, Germantown, MD, USA) was used for RNA isolation. RNA was obtained from cells cultured under aerobic and hypoxic conditions. cDNA synthesis on the mRNA template was performed using the RT2 HT First Strand Kit (Qiagen, Germantown, MD, USA). Expression of genes related to cell response to DNA damage was carried out using Real-time PCR using the RT$^2$ Profiler PCR Array—Human DNA Damage Signaling Pathway gene expression evaluation kit (Qiagen, Germantown, MD, USA). This kit allows the evaluation of the expression profile of 84 genes involved in DDR (Table 2).

**Table 2.** Genes analyzed with the RT$^2$ Profiler PCR Array—Human DNA Damage Signaling Pathway kit (Qiagen, Germantown, MD, USA).

|   | 1 | 2 | 3 | 4 | 5 | 6 | 7 | 8 | 9 | 10 | 11 | 12 |
|---|---|---|---|---|---|---|---|---|---|----|----|----|
| A | ABL1 | APEX1 | ATM | ATR | ATRIP | ATRX | BARD1 | BAX | BBC3 | BLM | BRCA1 | BRIP1 |
| B | CDC25A | CDC25C | CDK7 | CDKN1A | CHEK1 | CHEK2 | CIB1 | CRY1 | CSNK2A2 | DDB1 | DDB2 | DDIT3 |
| C | ERCC1 | ERCC2 | EXO1 | FANCA | FANCD2 | FANCG | FEN1 | GADD45A | GADD45G | H2AFX | HUS1 | LIG1 |
| D | MAPK12 | MBD4 | MCPH1 | MDC1 | MLH1 | MLH3 | MPG | MRE11A | MSH2 | MSH3 | NBN | NTHL1 |
| E | OGG1 | PARP1 | PCNA | PMS1 | PMS2 | PNKP | PPM1D | PPP1R15A | PRKDC | RAD1 | RAD17 |
| F | RAD21 | RAD50 | RAD51 | RAD51B | RAD9A | RBBP8 | REV1 | RNF168 | RNF8 | RPA1 | SIRT1 | SMC1A |
| G | SUMO1 | TOPBP1 | TP53 | TP53BP1 | TP73 | UNG | XPA | XPC | XRCC1 | XRCC2 | XRCC3 | XRCC6 |

*2.13. Data Analysis*

Data were expressed as means $\pm$ SEM from three experiments for viability tests. The IC$_{50}$ value was calculated from median effect plots for each of the compounds. Next, the corresponding dose for a given level of effect (i.e., percentage of affected/non-affected cells) was determined. The CI values were calculated from the IC$_{50}$ value. The theoretical basis of these calculations involves the median effect principle and was derived from the mass action law by Chou and Talalay. Reduction factor (Rf) was calculated as a ratio of the CI obtained for the drug combination treatment in cells untreated with a DRI to the CI obtained after pretreatment with each of the DRIs. For DNA damage and DNA repair analysis, the results were expressed as median; the lower and upper quartiles represent observations outside the $-9$ to 9 percentile range. Other results presented here are shown as a mean $\pm$ SD. Differences were evaluated with the Mann–Whitney U test. Data analysis was performed with Statistica software v. 13.3 (TIBCO Software Inc., Palo Alto, CA, USA).

### 3. Results

*3.1. Pretreatment with DRIs Enhances the Cytotoxic Effect of CDDP/VP-16 in Combined Treatment in Ovarian Cancer Cells at Normoxia Conditions*

We determined the effect of drugs (CDDP, VP-16) and DRI (YU238259, A12B4C3, DDRI-18) on the growth of CDDP-sensitive and CDDP-resistant cells belonging to experimental schemes 1 and 2 (ES 1 and ES 2). We also compared the ability of DRIs to sensitize ovarian cancer cell lines to CDDP and VP-16 used as a single drug and as a combined treatment. CDDP and VP-16 significantly decreased the viability of both cell lines in a concentration-dependent way on ES 1 (Figure S2A–D) and ES 2 (Figure S3A–D) cells. DRIs alone did not inhibit cellular growth in the range of concentrations up to 200 μM (Figure S4A–D). IC$_{50}$ values were calculated for all compounds using Compusyn software (Table 3). Cells were exposed to the indicated concentrations (Section 2) of the CDDP/VP-16 combination after pretreatment with DRI. The results of ES 1 (Supplementary) showed that DRIs are able to sensitize A2780 and A2780cis cells to CDDP/VP-16 (Tables S1 and S2). For ES 2 cells, the IC$_{50}$ values determined for the CDDP/VP-16 combinations were also compared with those obtained after the inclusion of HR or NHEJ inhibitors in the study. The introduction of DRI resulted in lower IC$_{50}$ values for the CDDP/VP-16 combination for each of the ES 2 cell lines (Table 4). We observed a significant effect of sensitization with YU238259 on cells sensitive to cisplatin and resistant to cisplatin, with Rf = 1.15 and 1.09, respectively. Additionally, A12B4C3 caused sensitization in both cell lines: Rf = 1.74 for PEA1 and Rf = 2.13 for PEA2 cells. DDRI-18 treatment resulted in over 2-fold sensitization for both ES2 cell lines (Table 4). As in the case of ES1 cells, the synergism of the CDDP/VP-16 action increased with increasing concentrations of these drugs. The introduction of NHEJ and HR inhibitors resulted in a decrease in the CI parameter, i.e., a change in the interaction between the drugs toward synergism (Table 5).

**Table 3.** IC$_{50}$ values determined under normoxia conditions for cell lines belonging to experimental schemes ES 1 and ES 2.

| Experimental Scheme | Cell Lines | IC$_{50}$ (μM)—Normoxia | |
|---|---|---|---|
| | | CDDP | VP-16 |
| ES 1 | A2780 | 13.35 | 34.48 |
| | A2780cis | 45.44 | 31.31 |
| ES 2 | PEA1 | 18.22 | 35.83 |
| | PEA2 | 35.96 | 31.27 |

**Table 4.** Comparison of IC50 and Rf values obtained for PEA1 and PEA2 cells after incubation with a combination of CDDP/VP-16 and DRIs. Rf > 1-sensitization.

| | | PEA1 | | PEA2 | |
|---|---|---|---|---|---|
| | | IC50 (μM) | Rf | IC50 (μM) | Rf |
| **Drugs** | CDDP/VP-16 | 9.49 | - | 11.13 | - |
| **Drugs + NHEJ inhibitor** | CDDP/VP-16/YU238259 | 8.25 | 1.15 | 10.21 | 1.09 |
| **Drugs + HR inhibitor** | CDDP/VP-16/ A12B4C3 | 5.45 | 1.74 | 5.22 | 2.13 |
| | CDDP/VP-16/ DDRI-18 | 4.65 | 2.04 | 4.38 | 2.54 |

**Table 5.** CI values obtained for PEA1 and PEA2 cells after incubation with CDDP/VP-16 and DRIs. CI values < 1 and CI > 1 indicate synergism and antagonism, respectively. Indications: ↓—decrease in CI values, ↑—increase in CI values.

| Drugs Concentration CDDP/VP-16 (μM) | CI (for Drugs) | CI Drugs + YU238259 | CI Drugs + A12B4C3 | CI Drugs + DDRI-18 |
|---|---|---|---|---|
| | | PEA1 | | |
| **6.75 (¹⁄8 + ¹⁄8 IC$_{50}$)** | 2.71 | ↓ 0.18 | ↓ 1.97 | ↓ 0.48 |
| **13.5 (¹⁄4 + ¹⁄4 IC$_{50}$)** | 2.43 | ↓ 0.41 | ↓ 1.65 | ↓ 0.17 |
| **27 (¹⁄2 + ¹⁄2 IC$_{50}$)** | 0.15 | ↑ 0.25 | ↓ 0.12 | ↓ 0.14 |
| **54 (1 + 1 IC$_{50}$)** | 0.65 | ↓ 0.18 | ↓ 0.23 | ↓ 0.23 |
| **108 (2 + 2 IC$_{50}$)** | 2.71 | ↓ 0.18 | ↓ 0.57 | ↓ 0.38 |
| | | PEA2 | | |
| **8.4 (¹⁄8 + ¹⁄8 IC$_{50}$)** | 2.95 | ↓ 0.77 | ↓ 0.26 | ↓ 0.14 |
| **16.7 (¹⁄4 + ¹⁄4 IC$_{50}$)** | 1.01 | ↓ 0.56 | ↑ 0.15 | ↓ 0.18 |
| **33.5 (¹⁄2 + ¹⁄2 IC$_{50}$)** | 1.05 | ↓ 1.04 | ↓ 0.37 | ↓ 0.26 |
| **67 (1 + 1 IC$_{50}$)** | 0.67 | ↑ 0.96 | ↑ 0.79 | ↓ 0.41 |
| **134 (2 + 2 IC$_{50}$)** | 2.95 | ↓ 0.77 | ↓ 0.26 | ↓ 0.14 |

*3.2. Pretreatment with DRIs Causes an Increase in the Level of DNA Damage and the Phosphorylated H2AX Accumulation in Ovarian Cancer Cells Treated with CDDP/VP-16 at Normoxia Conditions*

The DRIs caused a statistically significant ($p$ < 0.01) 5–10% increase in the level of DNA damage induced by the combination of CDDP/VP-16. An accumulation of DNA damage after pretreatment with A12B4C3 and DDRI-18 (Figure 3A,B and Figure S5A,B). The presence of DSB

was confirmed by significantly ($p < 0.05$) elevated levels ($p < 0.05$) of phosphorylated H2AX observed after introducing DRI into the treatment scheme (Figure 3C,D and Figure S5C,D).

### 3.3. Pretreatment with DRIs Decreases the Efficiency of DNA Damage Repair in Ovarian Cancer Cells Treated with CDDP/VP-16 at Normoxia Conditions

We observed a decrease in the level of drug-induced DSBs during repair incubation—a more than 10% difference in the level of DNA damage was noticed between the initial and final time points. DRI significantly slowed down the DNA repair in ES 1 (Figure S5E,F) and ES 2 (Figure 3E,F). The effect of sensitization was more pronounced for A12B4C3, with more than 5% differences in the level of DNA damage. Differences were observed between cells exposed only to CDDP/VP-16 compared to those additionally pretreated with A12B4C3 at the final time points.

### 3.4. Pretreatment with DRIs Causes an Accumulation of Cells at the G2/M Phase of the Cell Cycle in Ovarian Cancer Cells Treated with CDDP/VP-16 at Normoxia Conditions

Compared to the untreated control, cells accumulated at the G2/M phase. The introduction of DRIs to the scheme of therapy slightly changed the cell cycle distribution in ovarian cancer cells belonging to ES 1 (Figure S5G,H) and ES 2 (Figure 3G,H). About a 7–9% increase in the number of cells accumulated at the G2/M was noticed.

### 3.5. Pretreatment with DRIs Induce Apoptosis in Ovarian Cancer Cells Treated with CDDP/VP-16 at Normoxia Conditions

We observed that the combination CDDP/VP-16 caused an increase of over 100% in the caspase 3/7 activity. Furthermore, the introduction of DRIs caused a further significant ($p < 0.01$) increase ($p < 0.01$) of active caspases 3 and 7, especially among CDDP-resistant ovarian cancer cells. For cisplatin-sensitive variants, this was more than a 2-fold increase, and for cisplatin-resistant variants, it was more than a 10-fold increase. Additionally, this effect was not observed after introducing the caspase inhibitor Z-VAD-FMK into cells ES 1 (Figure S5I,J) and ES 2 (Figure 3I,J).

### 3.6. Cell Culture in Hypoxia Conditions Induces Increase in the Level of Transcription Factor Hif1A in Ovarian Cancer Cells

To confirm the presence of decreased oxygen concentration in cell culture, the level of transcription factor Hif1A was determined in lysates of ES 2 cells cultured under aerobic and hypoxic conditions. A statistically significant two-fold increase in the level of Hif1A was observed for cells cultured in a hypoxia environment compared to aerobic conditions (Figure S6).

### 3.7. Pretreatment with DRIs Does Not Enhance the Cytotoxic Effect of CDDP/VP-16 in Combined Treatment in Ovarian Cancer Cells at Hypoxia Conditions

In the case of cells belonging to ES 2, no enhancement of the cytotoxic effect of CDDP/VP-16 after introducing DRIs was observed under hypoxia, in contrast to normoxia conditions. DRIs did not sensitize ES 2 cells to the CDDP/VP-16 combination under hypoxia and did not increase the synergism of the CDDP/VP-16 combination effect (Table 6). Pretreatment with DRIs does not enhance the CDDP/VP-16 induced genotoxic effect, DNA repair, cell cycle distribution, and caspase level 3/7 in ovarian cancer cells at hypoxia conditions. We did not notice an effect of DRIs that was present under normoxia conditions. A higher level of endogenous damage was observed than under aerobic conditions—about 7% of DNA in the "comet tail". No statistically significant role of DRIs in increasing CDDP/VP-16-induced DNA damage levels was observed (Figure 4A,B). The use of DRIs in combination with CDDP/VP-16 did not increase the level of the phosphorylated form of histone H2AX in response to DNA damage in PEA1 and PEA2 cells (Figure 4C,D) and did not significantly affect DNA repair (Figure 4E,F). Furthermore, the combination did not significantly affect the percentage of U2 cells at the G2/M cell cycle checkpoint compared to unexposed controls. Exposure to the combination of CDDP/VP-16 and DRIs did not

result in changes in the percentage of cells arrested at the G2/M checkpoint compared to cells treated with CDDP/VP-16 without DRIs (Figure 4G,H). Although the combination resulted in a significant, more than threefold increase in the level of 3/7 caspases in the apoptosis pathway in the PEA2 cell line, the effect was not observed for PEA1 (Figure 4I,J).

**Table 6.** CI values obtained for PEA1 and PEA2 cells after incubation with CDDP/VP-16 and DRIs in hypoxia conditions. CI values < 1 and CI > 1 indicate synergism and antagonism, respectively. Indications: ↓—decrease in CI values, ↑—increase in CI values.

| Drugs Concentration CDDP/VP-16 (μM) | CI (for Drugs) | CI Drugs + YU238259 | CI Drugs + A12B4C3 | CI Drugs + DDRI-18 |
|---|---|---|---|---|
| | | PEA1 | | |
| 9.49 (¼ + ¼ IC$_{50}$) | 2.95 | ↓ 0.77 | ↓ 1.26 | ↑ 2.14 |
| 19 (½ + ½ IC$_{50}$) | 1.01 | ↑ 2.56 | ↑ 1.15 | ↑ 1.18 |
| 38 (1 + 1 IC$_{50}$) | 1.05 | ↑ 1.24 | ↑ 2.37 | ↑ 1.26 |
| 76 (2 + 2 IC$_{50}$) | 0.67 | ↑ 0.96 | ↑ 0.79 | ↓ 0.41 |
| | | PEA2 | | |
| 11.3 (¼ + ¼ IC$_{50}$) | 1.14 | ↑ 2.48 | ↑ 2.46 | ↑ 2.44 |
| 22.5 (½ + ½ IC$_{50}$) | 1.78 | ↑ 2.54 | ↓ 0.96 | ↑ 1.98 |
| 45 (1 + 1 IC$_{50}$) | 1.16 | ↑ 1.94 | ↑ 2.21 | ↑ 2.58 |
| 90 (2 + 2 IC$_{50}$) | 2.95 | ↑ 4.43 | ↑ 6.26 | ↑ 7.38 |

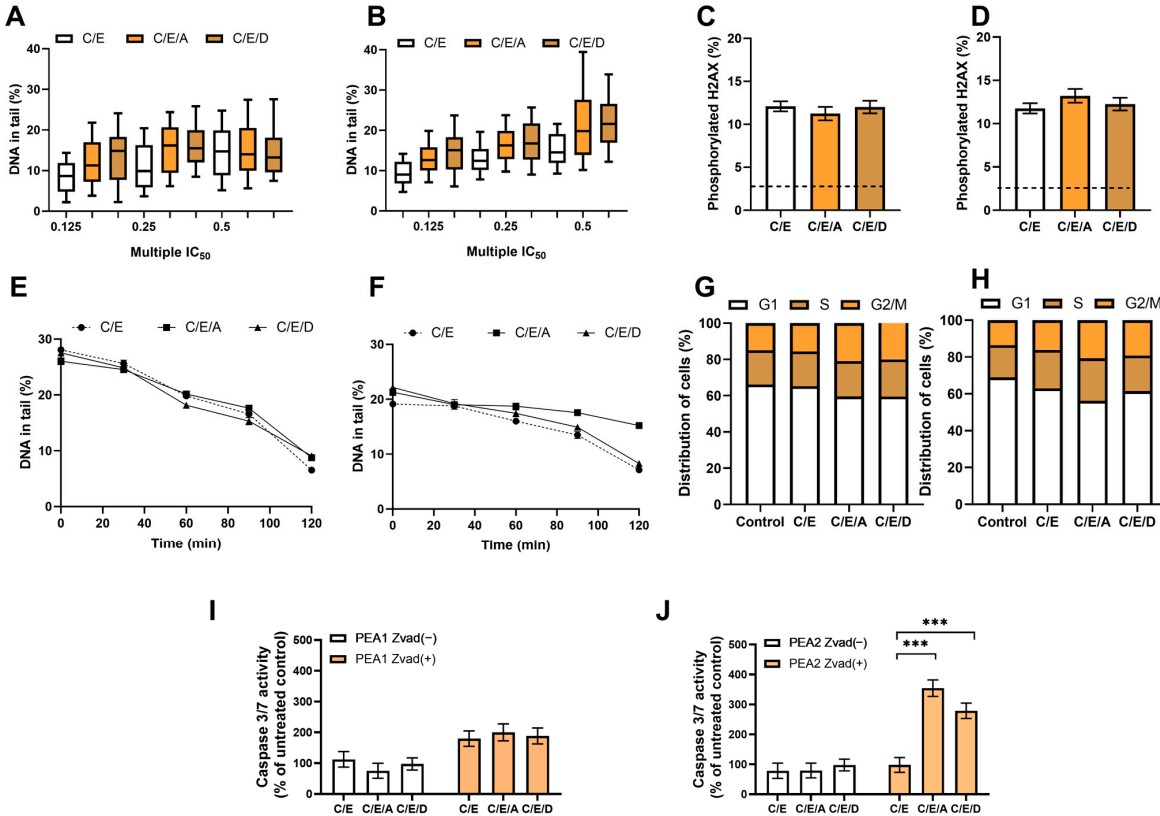

**Figure 4.** In hypoxia conditions, DRIs are unable to improve the activity of CDDP/VP-16 combination against human ovarian cancer cells PEA1 (**A,C,E,G,I**) and PEA2 (**B,D,F,H,J**). DRIs did not significantly

increase CDDP/VP-16 induced level of DNA damage (**A,B**), the level of phosphorylated H2AX (**C,D**), the DNA repair (**E,F**) and did not affect the distribution of cell cycle (**G,H**). A significant increase in the caspase 3/7 level was observed for CDDP-resistant PEA2 cells but not for its CDDP-sensitive counterpart (**I,J**). Results are presented as the mean ± SEM (**A,B**) or mean ± SD (**C–J**) of 3 independent experiments, ***—$p < 0.001$.

### 3.8. Gene Expression

Tables S1 and S2 present detailed data changes in the transcriptome levels of DDR genes under hypoxia conditions in ES 2 cells (Tables S3 and S4). Tables 7 and 8 summarize the results obtained for ES 2 cells. For the cisplatin-sensitive PEA1 line, a decrease in the expression levels of numerous genes involved in the apoptosis pathway, ATM/ATR signaling pathway, and DNA damage repair pathways was observed (Table 7). There was also a 2-3-fold decrease in the expression of several genes involved in these processes (Table S3). A large number of genes showed a decrease in expression in the case of cells of the PEA2 line. More than a 5-fold decrease in the expression of numerous ATM/ATR signaling pathway genes and DSB repair pathways was observed, and an increase in expression under hypoxia compared to aerobic conditions involved significantly fewer genes (Table 8). These results indicate a different profile of DDR under aerobic and hypoxic conditions both in the signaling and execution parts.

**Table 7.** Expression levels of genes involved in DDR in PEA1 cells under hypoxia conditions. Compared to control—expression levels under aerobic conditions. Green—decreased expression, orange—increased expression.

|   | 1 | 2 | 3 | 4 | 5 | 6 | 7 | 8 | 9 | 10 | 11 | 12 |
|---|---|---|---|---|---|---|---|---|---|----|----|----|
| A | ABL1 | APEX1 | ATM | ATR | ATRIP | ATRX | BARD1 | BAX | BBC3 | BLM | BRCA1 | BRIP1 |
| B | CDC25A | CDC25C | CDK7 | CDKN1A | CHEK1 | CHEK2 | CIB1 | CRY1 | CSNK2A2 | DDB1 | DDB2 | DDIT3 |
| C | ERCC1 | ERCC2 | EXO1 | FANCA | FANCD2 | FANCG | FEN1 | GADD45 | GADD45 | H2AFX | HUS1 | LIG1 |
| D | MAPK12 | MBD4 | MCPH1 | MDC1 | MLH1 | MLH3 | MPG | MRE11A | MSH2 | MSH3 | NBN | NTHL1 |
| E | OGG1 | PARP1 | PCNA | PMS1 | PMS2 | PNKP | PPM1D | PPP1R15A | PRKDC | PRKDC | RAD1 | RAD17 |
| F | RAD21 | RAD50 | RAD51 | RAD51B | RAD9A | RBBP8 | REV1 | RNF168 | RNF8 | RPA1 | SIRT1 | SMC1A |
| G | SUMO1 | TOPBP1 | TP53 | TP53BP1 | TP73 | UNG | XPA | XPC | XRCC1 | XRCC2 | XRCC3 | XRCC6 |

**Table 8.** Expression levels of genes involved in DDR in PEA2 cells under hypoxia conditions. Compared to control—expression levels under aerobic conditions. Green—decreased expression, orange—increased expression.

|   | 1 | 2 | 3 | 4 | 5 | 6 | 7 | 8 | 9 | 10 | 11 | 12 |
|---|---|---|---|---|---|---|---|---|---|----|----|----|
| A | ABL1 | APEX1 | ATM | ATR | ATRIP | ATRX | BARD1 | BAX | BBC3 | BLM | BRCA1 | BRIP1 |
| B | CDC25A | CDC25C | CDK7 | CDKN1A | CHEK1 | CHEK2 | CIB1 | CRY1 | CSNK2A | DDB1 | DDB2 | DDIT3 |
| C | ERCC1 | ERCC2 | EXO1 | FANCA | FANCD2 | FANCG | FEN1 | GADD45 | GADD45 | H2AFX | HUS1 | LIG1 |
| D | MAPK12 | MBD4 | MCPH1 | MDC1 | MLH1 | MLH3 | MPG | MRE11A | MSH2 | MSH3 | NBN | NTHL1 |
| E | OGG1 | PARP1 | PCNA | PMS1 | PMS2 | PNKP | PPM1D | PPP1R15 | PRKDC | PRKDC | RAD1 | RAD17 |
| F | RAD21 | RAD50 | RAD51 | RAD51B | RAD9A | RBBP8 | REV1 | RNF168 | RNF8 | RPA1 | SIRT1 | SMC1A |
| G | SUMO1 | TOPBP1 | TP53 | TP53BP1 | TP73 | UNG | XPA | XPC | XRCC1 | XRCC2 | XRCC3 | XRCC6 |

## 4. Discussion

Acquisition of resistance to anticancer drugs, especially platinum-based compounds, is currently one of the biggest challenges in the development of new treatment schemes for OC. In the present work, we decided to use three DRIs as molecules that can potentially sensitize EOC cells to anticancer drugs (CDDP and VP-16) by inhibiting the DSB repair pathways.

DRIs as potential anticancer drugs have been tested in clinical trials and introduced into the treatment scheme of some particular cancers [23]. The most striking example is PARP inhibitors; however, they are limited to killing cancer cells that show a BRCAness phenotype and are not able to act successfully in other types of cancer [24,25].

Our study was carried out on a panel of four ovarian cancer cell lines. Some of the limitations of A2780/A2780cis cells were mentioned in Section 2. Another limitation of these cells is the unclear origin of A2870. The most common type of ovarian cancer is serous HGSOC, which also has the worst prognosis. Therefore, this histologic type should be used for basic and preclinical research. Unfortunately, the cell lines A2780, SKOV, OVCAR3, CAOV3, and IGROV1, most commonly used for ovarian cancer research [26,27], are not ideal models for HGSOC. Additionally, the A2780 line has been described as isolated from endometrioid adenocarcinoma, and further studies have confirmed this classification. These cell lines do not exhibit the main characteristics of HGSOC, such as high levels of CNV (Copy Number Variation) and the presence of mutations in the TP53, BRCA1, and/or BRCA2 genes. Instead, mutations are present in genes other than HGSOC, such as ARID1A (characteristic of endometrioid and clear cell ovarian cancer) and PIK3CA (identified in clear cell ovarian cancer) [26,28]. However, these cell lines are well described in the literature, have long research, and are characterized by high cell viability, low culture requirements, and rapid division.

Given the above limitations of the ES 1 model, two more cell lines, PEA1 and PEA2 (ES 2 model), were included in the study after verifying the hypothesis on this model. ES 2 consists of the cisplatin-sensitive variant derived from cells obtained from patients before treatment and the cisplatin-resistant variant obtained after chemotherapy with platinum compounds and after the development of clinical resistance to CDDP [19].

We observed that introducing DRIs that are different than PARP inhibitors to the treatment scheme is beneficial. It decreased the effective dose of anticancer drugs used alone or in combination in CDDP-sensitive (A2780 and PEA1) and its CDDP-resistant counterparts (A2780cis and PEA2 cells), but not on PEO1/PEO4 cells. We obtained positive results for the DRIs tested, but the effect of sensitization was more pronounced for the NHEJ inhibitors.

The mechanism of action of drugs used in this study: CDDP and VP-16 involve the formation of DSBs directly and indirectly. VP-16 induces DSB directly by stabilizing the cleavable complex of Top2, which leads to the cessation of the religation of DNA during the entire cell cycle [29,30]. CDDP interacts with DNA via the formation of intrastrand and interstrand adducts, which can be converted into disiloxane (DSB) during the process of the cross-link repair process. Both types of DNA lesions are classified as replicative DNA lesions. They arise during replication when replicative machinery meets obstacles like the cleavable complex of Top2 or intra- and interstrand adducts during replication progress. Generate replication stress. In response to replication stress, eukaryotic cells mainly trigger HRR, but also NHEJ [31,32].

The HRR pathway is considered one of the crucial components involved in the development of resistance to anticancer drugs [33]. In the early stages of OBE, defects in the HR pathway (decreased expression of BRCA1 or FANCF, mutations in BRCA1 and BRCA2) can contribute to the sensitivity of OBE to platinum-based treatment [34,35]. However, in the advanced stages of OCE, genetic reversion of BRCA1/2 mutations and reexpression of FANCF can be correlated with increased resistance to platinum-based anticancer drugs [36,37]. It was also demonstrated that high levels of HR-involved RAD51 protein may induce an increase in VP-16-induced DSB repair [38,39]. Due to the crucial role of DSBs DNA repair in the development of drug resistance in EOC, introducing the DSBs inhibitors into the treatment scheme seems to be reasonable.

According to recent data, a couple of DNA repair inhibitors have been tested as potential sensitizers for ovarian cancer cells together with commonly used anticancer drugs. Platinum-based agents (CDDP and carboplatin) were combined with inhibitors of the TLS, NER, and HRR pathways and VP-16—with NHEJ inhibitors (especially interacting with

DNA PKc and ligase IV) [40]. However, for now, the only one introduced to clinical use was PARP inhibitor-olaparib [41].

The combination of DRIs with CDDP/VP-16 enhanced cytotoxicity and genotoxicity in ES 1 and ES 2 cells. The best synergistic effect occurred when DRIs were added first to induce NHEJ or HRR deficiency. These data are consistent with other reports where DRIs were administered prior to other drugs to enhance their cytotoxic effect [42–45].

The scheme of combining DRIs with chemotherapeutics or radiation therapy was based on several hours of preincubation with DRIs in most of the published works. Experiments for the present study were designed similarly. An experimental scheme based on preincubation with compounds that affect DNA repair pathways or DNA molecules is also frequently used. Blagosklonny et al., in their study, used HCT116 colon cancer cells and a variant of them with defects in cell cycle checkpoint function. They showed that preincubation with doxorubicin, a drug that interacts with DNA, enhances the anticancer activity of paclitaxel, vinblastine, and epotilones [46]. Toulany et al. conducted studies on the A549 human lung cancer cell line with a KRAS mutation. They showed that the use of an inhibitor that acts on the epidermal growth factor receptor (EGFR) before radiotherapy inclusion affects the inhibition of the EGFR-dependent PI3K/AKT pathway (phosphatidylinositol 3-kinase/serine/threonine Kinase 1) pathway. The consequence is a reduced level of DNA-PKcs activation, which results in a reduced ability to repair DSB and limits cell survival after irradiation [47]. Chen and Yu described the possibility of using COH34, a poly(ADP-ribose) glycohydrolase inhibitor, as a potentiator of CDDP, doxorubicin, temozolomide, and camptothecin against a wide panel of cell lines representing various types of human cancers, including PEO1/4 [48].

In hypoxia, a reduction in the positive effect of NHEJ inhibitors was observed compared to the results obtained for normoxia conditions. In the case of HR inhibitors, even a decrease in the cytotoxicity of the CDDP/VP-16 combination was observed when combined with DRI compared to the results obtained for the drugs alone. Therefore, changes in the expression levels of 84 genes involved in the signaling pathways of the cell response to DNA damage were analyzed. The results were compared with those obtained for normoxia conditions. The absence of oxygen in the environment modulates the expression of genes that play a role in the cellular response to DNA damage. These changes included a 2-fold reduction in the expression levels of genes involved in signaling pathways involved in ATM/ATR kinase activity and apoptosis, as well as a 2- or 3-fold increase in the expression of genes involved in DNA repair in the cisplatin-sensitive PEA1 line.

In the cisplatin-resistant PEA2 cell line, we observed changes in the expression levels of more genes compared to the cisplatin-sensitive counterpart. These changes focused mainly on genes involved in ATM/ATR signaling pathways and DNA double-strand break repair pathways. A more than five-fold decrease in the expression of genes involved in DNA double-strand break repair (including RAD51 AND XRCC2) was observed. This suggests that hypoxia remodulates DDR to a very high degree, which affects the overall response of the cell to DNA-damaging drugs. This is not surprising, given the fact that there is a rather strong relationship between the HIF1A protein and the kinases PI3KK, ATM, ATR, and DNA-PK. The absence of ATM protein is correlated with increased expression and activity of the HIF1A protein [49,50]. Similarly, ATM inhibition in a mouse model resulted in inhibition of senescence induction and increased tumor size and invasiveness [51]. Under hypoxic conditions, ATM phosphorylates the HIF1A protein, which increases the stability of HIF1A, probably through post-translational modifications. ATM, in combination with several other factors, is also involved in maintaining elevated mTORC1 (Mechanistic Target of Rapamycin Complex 1) activity in hypoxic tumors [52,53].

The concept of using DRIs in cancer therapy has proven its usefulness and is already being successfully applied, with a measurable result being the introduction of PARP inhibitors into clinical practice. However, the use of PARP inhibitors is most relevant for the treatment of cancers characterized by the presence of mutations within the BRCA1 and BRCA2 genes. The occurrence of these mutations is not a characteristic of all ovarian

malignancies, so the search for new DRIs that work effectively in tumors with a different genetic profile is justified. In the classical and alternative cases of ovarian cancer, the NHEJ repair pathway is a good target for the use of inhibitors, and the preliminary results obtained in this study can serve as a starting point for the search for new, therapeutically effective DRIs. NHEJ inhibition not only sensitizes cancer cells to DSBs-inducing compounds but also disrupts HIF-dependent regulation. It is the hypothesis that the use of NHEJ inhibitors alone can induce cell death via synthetic lethality, as DDR is altered under hypoxia, and these changes are exacerbated in some ovarian cancers with mutated MMR pathways. Additionally important is the fact that ovarian cancer cells cultured under normoxia and hypoxia respond differently. This is important because one of the determinants of drug resistance of tumor cells (including CDDP resistance in the case of ovarian cancer) is the hypoxia conditions prevailing in the tumor tissue environment. Hypoxia is defined as a state of oxygen deficiency in the environment. It occurs as a result of a decrease in the partial pressure of oxygen (pO2) in the environment or under conditions of inadequate oxygen supply [54]. Lower physiological oxygen levels are a common phenomenon in malignancies. Cancer cells are characterized by a higher proliferation rate compared to normal cells. Therefore, oxygen demand for oxygen is higher than the ability to deliver it to the tissue. Abnormal angiogenesis in tumorigenesis also contributes to inadequate oxygen supply to tumor tissue [54,55]. Decreased oxygen levels are associated with several metabolic changes in the cell. In response to reduced oxygen supply, the transcription factor HIF1A is activated. Other signaling pathways responsible for proliferation, apoptosis, and metabolism are also disrupted. These include PI3K/AKT/mTOR, MAPK, and NFB [54,56]. In this context, it is reasonable to ask about the desirability of the idea of using DRIs in cancer therapy since ovarian cancers are characterized by hypoxia and different DDRs. Tumor cells are characterized by heterogeneity in terms of oxygen availability. Perfusion-limited hypoxia occurs transiently, with inadequate oxygen delivery due to malfunctioning blood vessels subjected to repeated cycles of closure and reopening. This results in slowed blood flow through the vessels and changes in the supply of oxygen to tumor cells. These constant changes lead to cyclic periods of hypoxia and reoxygenation, resulting in the development of a heterogeneous population of cells within the tumor [57,58]. Diffusion-limited hypoxia (also known as chronic or persistent hypoxia) is another type of tumor hypoxia and refers to the permanent restriction of oxygen diffusion via an abnormal vascular network. Chronic hypoxia occurs when tumor cells grow more than 70 μm from preexisting blood vessel networks, supplying them with blood and preventing adequate oxygen delivery [59]. Therefore, DRI therapy will be effective against some tumor cells. In addition, a multipronged approach to cancer therapy is now favored. One direction to support classical therapy is the reoxygenation of solid tumors to counteract hypoxia. Several different strategies have been proposed; among them, most studies have focused on HAPs—hypoxia-activated prodrugs (HAPs), inhibition of HIF signaling, targeting important hypoxia pathways such as UPR (Unfolded Protein Response) and mTOR, and metabolic interventions [60]. Such approaches would enable the regimen proposed in this work to effectively kill cancer cells.

## 5. Conclusions

Overall, these results suggest that the DNA double-strand break repair inhibitors used in this study can be useful as agents to overcome the cisplatin resistance in human ovarian cancer cells; however, this action is mainly limited to normoxic conditions as ovarian cancer cells change their cellular biochemistry related to DDR in hypoxia.

**Supplementary Materials:** The supporting information can be downloaded at: https://www.mdpi.com/article/10.3390/cimb45100500/s1.

**Author Contributions:** Conceptualization, A.M. and T.P.; methodology, A.M. and T.P.; software, A.M.; validation, A.M. and T.P.; formal analysis, A.M. and T.P.; investigation, A.M., I.G. and T.P.; resources, A.M.; data curation, A.M.; writing—original draft preparation, A.M. and T.P.; writing—review

and editing, A.M. and T.P.; visualization, A.M. and T.P.; supervision, T.P.; project administration, A.M.; funding acquisition, A.M. All authors have read and agreed to the published version of the manuscript.

**Funding:** This research was funded by the National Science Centre (Poland), grant number according to decision No. DEC-2016/23/N/NZ7/02023.

**Institutional Review Board Statement:** Not applicable.

**Informed Consent Statement:** Not applicable.

**Data Availability Statement:** All data is available from A.M. on reasonable request.

**Acknowledgments:** This research was funded by the National Science Centre (Poland), grant number UMO-2016/23/N/NZ7/02023.

**Conflicts of Interest:** The authors declare no conflict of interest. The funders had no role in the design of the study; in the collection, analyses, or interpretation of data; in the writing of the manuscript; or in the decision to publish the results.

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
