# Peer review of "DNA Double-Strand Break Repair Inhibitors: YU238259, A12B4C3 and DDRI-18 Overcome the Cisplatin Resistance in Human Ovarian Cancer Cells, but Not under Hypoxia Conditions"

_cimb, doi:10.3390/cimb45100500_

Round 1

Reviewer 1 Report

In this study, a new approach to sensitize CDDP resistant, human ovarian carcinoma cells to combined treatment with CDDP and VP-16 was presented.The researchers stated that all of inhibitors enhanced therapeutic effect of CDDP/VP-16 treatment scheme and allowed to decrease the effective dose of CDDP/VP16. And also, the researcher were found that  inhibition of HRR or NHEJ decreased survival and increased DNA-damage level, increased amount of γ-H2AX foci and caused an increase of apoptotic fraction after CDDP/VP16 treatment. In the result of study,  a delayed repair of DSBs was detected in HRR or NHEJ inhibited cells were determinated.

The materials selected and the methods applied by the researchers to realise the stated aim were appropriate and adequate, and detailed information about the methods was given. The results obtained in the study were given in detail, 4 figures and 8 tables were added to make the findings more understandable. The researchers compared their findings with the data in many sources related to the subject of the study and discussed the method they applied for the human ovarian cancer cells.

Author Response

Thank you for your review and appreciation of the effort in writing this manuscript.

Reviewer 2 Report

The article "DNA double-strand break repair inhibitors: YU238259, A12B4C3 and DDRI-18 overcome the cisplatin resistance in human ovarian cancer cells, but not under hypoxia conditions" is an interesting scientific study.            The article is prepared in a standard way, and all its subsections are prepared synthetically and correctly. The Material and Methods section is prepared in detail. It is worth emphasizing the Authors' explanation of why they used these cell lines for their research.                                                                   The use of multiple research methods is also very important.                     The Authors could attach a graphic abstract, which is a synthetic description of the research conducted.                                                                           Due to the scientific aspect of the obtained results and the potential possibility of use in clinical trials, I suggest printing the article "DNA double-strand break repair inhibitors: YU238259, A12B4C3 and DDRI-18 overcome the cisplatin resistance in human ovarian cancer cells, but not under hypoxia conditions" in CIMB in its current form.

Author Response

Thank you for your review and appreciation of the effort we put into our manuscript. We noticed that none of the manuscripts published in CIMB have graphic abstracts. We shall, obviously, include one if the editor considers it appropriate.

Reviewer 3 Report

In this study, Macieja et al. demonstrated a novel approach to sensitize CDDP-resistant ovarian carcinoma cells by introducing DNA double strand break (DSB) repair inhibitors (HRR and NHEJ inhibitors) to CDDP/VP-16 treatment, enhancing therapeutic efficacy and reducing the required dosage. However, they noted that hypoxia conditions altered the transcriptome, potentially activating alternative DSB repair systems, which affected treatment response. This study highlights the potential of targeting HRR and NHEJ pathways to overcome CDDP resistance in ovarian cancer, while noticing the impact of hypoxia on treatment outcomes.

The study is interesting. However, please check some minor comments:

1) Kindly breakdown the Introduction section with multiple paragraphs. 

2) Add a novelty statement at the end of the Introduction.

3) Please microscopic images of the cell lines.

4) Add a concise conclusion. 

5) Please cite recent references (last 3-4 years), at least for some general statements in Introduction.     

Minor editing of English language required

Author Response

1) Kindly breakdown the Introduction section with multiple paragraphs. 
We have done so.

2) Add a novelty statement at the end of the Introduction.
A novelty statement at the end of the Introduction was added

3) Please microscopic images of the cell lines.
Microscopic images of the cell lines used in this study were included in the supplement section.

4) Add a concise conclusion. 
We have added a concise conclusion

5) Please cite recent references (last 3-4 years), at least for some general statements in Introduction.
Novel referenced were added
The manuscript was also checked for language errors by a linguist
Thank you for your valuable comments